

# Effect of *Bacillus subtilis* on antioxidant enzyme activities in tomato grafting

Maria D. Arias Padró[1], Emilia Caboni[2], Karla Azucena Salazar Morin[1], Marco Antonio Meraz Mercado[1] and Víctor Olalde-Portugal[1]

[1] Departamento de Biotecnología y Bioquímica, Centro de Investigación y de Estudios Avanzados del IPN, Irapuato, Guanajuato, Mexico
[2] Consiglio per la Ricerca in Agricoltura e l'Analisi dell'Economia Agraria (CREA), Olivicoltura, Frutticoltura e Agrumicoltura (OFA), Rome, Italy

Corresponding author
Víctor Olalde-Portugal,
victor.olalde@cinvestav.mx,
v_olalde@yahoo.com.mx

## ABSTRACT

Grafting generally means stress to a plant and this triggers antioxidant defense systems. An imbalance in reactive oxygen species may negatively affect the grafting success. Several research projects have studied the association with plant growth-promoting rhizobacteria (PGPR) and it has been documented that they enhance nutrient acquisition, regulate hormone levels, and influence the antioxidant response in crops. However, little is known about the strategy of inoculating grafted herbaceous plants with PGPR and its effect on the antioxidant response. The effects of inoculating a strain of *Bacillus subtilis* on the antioxidant metabolism of grafted tomato were evaluated. In this study, two different rootstocks were used for tomato (*Solanum lycopersicum* L. var. Rio Grande (RG)): [*S. lycopersicum* L. var. cerasiforme (Ch)] and eggplant [(*Solanum melanogena* L. (Ber)] to establish a compatible graft (RGCh) and a semicompatible graft (RGBer). Enzyme activities involved in the antioxidant defense system: superoxide dismutase (SOD), catalase (CAT), phenylalanine ammonia lyase (PAL), polyphenol oxidase (PPO), peroxidase (POD), and total phenols were measured during 4 weeks after grafting. The results show that for RGCh, during the first two weeks after grafting, the tendency was a decrease of the enzyme activity for SOD, CAT, PAL when inoculated with *B. subtilis*; while in the semicompatible graft RGBer, PPO and PAL decreased their activity after inoculation. For both combinations, the quantity of total phenols varied depending on the day. In both graft combinations, applications of *B. subtilis* resulted in 86 and 80% callusing compared with the uninoculated control where the percentages were 74 and 70% for RGCh and RGBer, respectively. The highest significant graft success (95%) was recorded 28 days after grafting for inoculated RGBer. These findings imply that *B. subtilis* induced antioxidant mechanisms in grafted plants and suggest that inoculation with this growth-promoting bacterium can represent a biotechnological approach to improve success in tomato grafting.

## INTRODUCTION

Grafting is a horticultural technique that has been practiced since ancient times (*Mudge et al., 2009*). It is very important for woody plants but, in the last century, grafting has become important in the Cucurbitaceae (i.e., watermelon, melon, cucumber) and

Solanaceae family (i.e., tomato, eggplant, and pepper) (*Bletsos & Olympios, 2008*). Grafting is also widely used in tomatoes to confer resistance to biotic and abiotic stresses (*Singh et al., 2017*). Successful grafting may be influenced by factors such as time of grafting, hormonal application, compatibility of the species (*Gainza, Opazo & Muñoz, 2015*), as well as level of the mechanical damage. The latter factor, in particular, can generate an antioxidant response due to the formation of ROS (Reactive Oxygen Species) (*Suzuki & Mittler, 2012*). Among them, superoxide radical ($O^{-2}$) and hydroxyl radical ($OH^{-}$) are free radicals that can adversely alter DNA, protein, lipids and carbohydrates or activation and inactivation of enzymes which can lead to cell damage and death (*Baxter, Mittler & Suzuki, 2014*). Therefore, the control of tissue damage and, consequently, the success of the grafting may be related to variation in the activity of enzymes or content of other non-enzymatic molecules related to the antioxidant metabolism. The enzymes superoxide dismutase (SOD), catalase (CAT) and peroxidase (POD) can be biochemical markers of oxidative damage and the level of activity could be a sign of resistance to stress (*Gill & Tuteja, 2010*; *Maksimović et al., 2013*). However, there are other non-proteic substances, such as polyphenols, that are involved in the scavenging of ROS (*Foyer & Noctor, 2013*). Phenolic compounds are products of the secondary metabolism of plants. The enzymes polyphenoloxidase (PPO) and POD, are involved in the oxidation of phenolic compounds, catalyzing the oxidation of phenols into quinones, which can spontaneously polymerize to form dark pigments (*Constabel & Barbehenn, 2008*). POD, PPO as well as phenylalanine ammonia lyase (PAL), the first enzyme involved in the phenylpropanoid pathway and, therefore in the biosynthesis of the polyphenol compounds, also play a relevant role in plant resistance to stress (*Finger, 1994*; *Soares et al., 2005*).

A considerable number of bacterial species, mostly associated with the plant rhizosphere, have been tested and found to be beneficial for plant growth, yield and crop quality. They have been called "plant growth-promoting rhizobacteria" (PGPR) and include strains of the genus *Bacillus* (*Rodriguez & Fraga, 1999*; *Sturz & Nowak, 2000*; *Sudhakar et al., 2000*; *Ruzzi & Aroca, 2015*). Microbial cells have several antioxidant defense mechanisms. *Bacillus* species and many other bacteria exert antioxidant activity producing a range of enzymes (*Kaizu et al., 1993*; *Ahotupa, Saxelin & Korpela, 1996*; *Amanatidou et al., 2001*; *Lin & Chang, 2000*). Among these, *Bacillus subtilis* produces two CATs (*Loewen & Switala, 1987*) and SOD (*Murphy et al., 1987*) as well as other antioxidant metabolites (*Kaspar, Neubauer & Gimpel, 2019*). In addition, *Bacillus* spp. induce in colonized plants antioxidant enzymes, such as SOD, CAT, POD, PPO, PAL, and phenolic acids favoring plant response to stress conditions (*Radhakrishnan, Hashem & Abd Allah, 2017*; *Rais et al., 2017*). The positive impact of *B. subtilis* has also been shown in tomato plants in biocontrol of bacterial wilt caused by *Ralstonia solanacearum*, through a role in increasing activities of PAL, PPO, POD, and SOD (*Li et al., 2008*), as well as in growth stimulation and induction of systemic resistance in tomato against early and late blight by inducing defense-related enzymes such as PPO, POD, and SOD (*Chowdappa et al., 2013*).

Recent reports suggest that grafting tomato varieties onto suitable *Solanum* spp. rootstocks can alleviate damage caused by soilborne pathogens and the adverse effects of abiotic stresses besides enhancing the efficiency of water and nutrient use of plants (*Singh*

*et al., 2017*). In addition, grafting tomato on eggplant is a potential tool for improving waterlogging tolerance and related resistance to tomato bacterial wilt disease (*Bahadur et al., 2015*; *Kariada & Aribawa, 2017*).

Thus, this study was focused on defining the effects of the PGPR, *B. subtilis*, on grafting of tomato plants on tomato (compatible rootstock) and eggplant (semicompatible rootstock) by assessing modifications in the activity of SOD, CAT, PAL, POD, PPO, and in phenol content. To the aim, a preliminary in vitro antioxidant activity of *B. subtilis* was performed, then the scions of a tomato variety were immersed into the bacterial solution of *B. subtilis* and grafted on the rootstocks. Then activities of enzymes, SOD, CAT, PAL, POD, PPO, total phenols were measured during 4 weeks after grafting, to characterize their involvement in the antioxidant defense system during the plant response to the grafting process.

## MATERIALS & METHODS

### Preparation of inoculum of strains

Eight strains of *B. subtilis* provided by Biotecnología Microbiana S.A. de C.V. were used. Inoculum of the eight strains was prepared for all experiments by harvesting cells from cultures previously grown on potato dextrose (PD) broth at 28 °C for 24 h on an orbital shaker at 150 rpm. The concentration of the inoculum was adjusted to $10^6$ CFU/mL $\approx 0.1$ $OD_{535}$ nm using a spectrophotometer reading (*Thompson, Clarke & Kobayashi, 1996*).

### In vitro antioxidant activity of *B. subtilis*
#### Resistance to hydrogen peroxide ($H_2O_2$)

The method of *Kadaikunnan et al. (2015)* was used with some modifications. $10^6$ CFU/mL $\approx 0.1$ $OD_{535}$ nm of strains of *Bacillus* cells were grown in 500 mL Erlenmeyer flasks containing 250 mL PD broth supplemented with 0.2, 0.4, 0.6, 0.8, or 1 mM $H_2O_2$ at 28 °C on an orbital shaker at 150 rpm for 24 h. The control treatment consisted of the growing medium inoculated with *B. subtilis* hydrogen peroxide-free. Cell growth was measured spectrophotometrically at 535 nm every hour and increases in cell growth were measured as increases in optical density (OD).

#### Hydroxyl radical scavenging activity (OH·)

Once the strain growth corresponding to $10^6$ CFU/mL $\approx 0.1$ $OD_{535}$ nm was achieved, neutralization of the hydroxyl radicals was determined using the Fenton reaction, according to *Kadaikunnan et al. (2015)*. Briefly, one mL of bright green reagent (0.435 mM), two mL $FeSO_4$ (0.5 mM) and 1.5 mL $H_2O_2$ (3% w/v) were mixed with different volumes of each strain (0.5, 1.0, 1.5, 2.0 and 2.5 mL). The suspensions were incubated at room temperature for 15 min, and then the absorbance was spectrophotometrically measured at 624 nm. The ability of the bacteria to scavenge hydroxyl radicals was determined according to the following equation.

$$Scavenging\ activity\ (\%) = \left[ \frac{(A_s - A_0)}{(A - A_0)} \right] \times 100$$

where, $A_s$ is the absorbance of the sample, $A_0$ is the absorbance of the control in the absence of the sample, and A is the absorbance without the sample and the Fenton reaction system.

The change in the absorbance of the reaction mixture indicated the scavenging ability of *B. subtilis* for hydroxyl radicals.

### *Total antioxidant activity (DPPH free radical scavenging activity)*

The total antioxidant activity (TAC) of *B. subtilis* strains was evaluated by the method described by *Kadaikunnan et al. (2015)*. Once an OD of 0.1 ($10^6$ CFU/mL) of *B. subtilis* cells at 535 nm was obtained, 0.5, 1.0, 1.5, 2.0 and 2.5 mL of the bacterial cells were mixed with one mL of the DPPH (1,1-diphenyl-2-picrylhydrazyl) solution (0.05 mM). The mixture was stirred and incubated in the dark for 30 min at room temperature. The controls were deionized water and DPPH solution and the blanks contained only methanol and bacterial cells. The absorbance of the solution was measured at 517 nm after centrifugation of the samples at 16,218 g for 10 min. TAC was determined by the following equation:

$$Total\ antioxidant\ activity\,(\%) = \left[1 - \frac{(A_{sample} - A_{blank})}{A_{control}}\right] \times 100$$

where $A_{sample}$ is the absorbance of the sample, $A_{blank}$ is the absorbance of methanol with bacterial cells and $A_{control}$ is the absorbance of deionized water and DPPH reagent (*Brand-Williams, Cuvelier & Berset, 1995*).

## Plant material

*Solanum lycopersicum* L. (tomato, var. Rio Grande and var. cerasiforme) and *Solanum melongena* L. (eggplant) seedlings were grown in the experimental greenhouse of the Ecological Biochemistry Laboratory at CINVESTAV (Advanced Research Center of the National Polytechnic Institute, Irapuato, Guanajuato, Mexico).

The commercial var. Rio Grande® was used as scion. This is one of the industrial varieties mostly cultivated in Mexico. The tomato "Cherry" and the eggplant were used as rootstocks. The choice of the rootstocks was made based on the degree of compatibility and spectrum of resistance/tolerance to biotic adversities.

The plants were germinated in trays containing a mixture of lime, vermiculite, perlite, leaf mold, and Sunshine® Mixture no. 3 (1:1:1:2:3). When the seedlings had developed four or five true leaves in the case of tomato and two or three true leaves in the case of eggplant, after 30 and 40 days of growth, respectively (late spring), plants were used for grafting: tomato, var. Rio Grande, was grafted on tomato var. cerasiforme (RGCh) and eggplant (RGBer).

## Plant inoculation and grafting

*Bacillus subtilis* strain BMB 44 was prepared for inoculation by harvesting cells from PD broth cultures grown at 28 °C for 24 h on an orbital shaker at 150 rpm. The concentration of the inoculum was adjusted using a spectrophotometer to $10^6$ CFU/mL $\approx$ 0.1 OD$_{535}$ nm (*Thompson, Clarke & Kobayashi, 1996*). The adopted strain was chosen among those preliminary tested and giving the highest antioxidant response.

The seedlings chosen for grafting had all the same diameter (1.5–2.0 mm). The graft cut was made with a half-size double-edge razor blade. The splice grafting technique was used: the rootstock was cut at a 45 °C angle above the cotyledons and the scion was cut at the same angle as the rootstock. The height of the rootstock and the scion were 2.0–2.5 cm and 4–4.5 cm, respectively. After cutting the scion parts, inoculation was performed by immediately immersing one cm of the basal part of the scion in bacteria suspension (or tap water for the control) and incubated at room temperature for 15 min. After treatments,

grafting was immediately performed and both parts of the plants were held with a silicone grafting clip.

For this study a completely randomized design was used: 2 (treatments) × 3 (analyses times) × 2 (grafting combinations). Each treatment was repeated three times (replicates). Each replicate consisted of 30 grafted plants.

## Post-graft plant healing and grafting success rate

The post-grafting healing was performed in containers (23 × 15 × 14.5 cm (L × W × H)) with a plastic dome. Each container had 30 plants. Recovery was carried out in the growing chamber of the Department of Biotechnology and Biochemistry of CINVESTAV. The conditions of the growing chamber were $25 \pm 1\,°C$ with a photoperiod of 16 h, 117 µmol $s^{-1}\,m^{-2}$, and relative humidity was between 85–95%, according to the humidity data logger. Seven days after grafting (DAG) the seedlings were irrigated again but the dome was partially opened to gradually reduce humidity up to 70%. Plantlets were kept under these conditions for 28 days. Rootstock lateral suckers were removed by hand when necessary.

To evaluate the relationship between *B. subtilis* and graft success, the level of callus development around the graft union was determined 4, 8, and 15 days after grafting. The grafted tomatoes were grouped in plants with a completely developed callus, partially developed callus, or absence of callus. The overall evaluation of grafting success was performed 28 days after grafting.

To determine the presence of *B. subtilis* in grafted plants, a preliminary experiment was conducted according to *Falcão et al. (2014)*. Fragments of leaves, stems (scion and rootstocks) and shoot apexes from 1, 15, and 28-day-old inoculated grafted plants were used. Non-inoculated grafted plants were used as control.

## Sample collection and enzyme extraction

To evaluate enzyme (SOD, CAT, POD, and PPO) activities and total phenol content, stem sections of the treated and control plants were used. For each treatment, the samples were represented by two mm stem sections (1 mm above and one mm below the grafting point) per replication. These were collected 1, 15, and 28 DAG. Collected samples were frozen in liquid nitrogen and stored at - 80 °C for subsequent determination of enzyme activity and total phenol content. The method described by *Giannopolitis & Ries (1977)* was adopted for the extraction of antioxidant enzymes. The stem samples (0.1 g) were homogenized in pre-chilled pestle and mortar. 450 µL of ice-cold 50 mM phosphate buffer, pH 7.0, and 50 µL of 10 mM EDTA solution (1:5 w/v) were added to the homogenate and centrifuged at 18,000 g at 4 °C for 15 min. The supernatants were immediately used for the determination of the activities of the enzymes. All steps in the enzyme extraction were carried out at 0−4 °C. Enzyme activities were expressed as U/mg protein.

The method described by *Beaudoin-Eagan & Thorpe (1985)* was used for the extraction of enzyme PAL. The stem samples (0.1 g) were homogenized in pre-chilled pestle and mortar. 200 µL of ice-cold and 0.5M Tris-HCl, pH 8, (1:2 w/v) were added to the homogenized sample and centrifuged at 15,000 g at 4 °C for 10 min. Enzyme activity was expressed as U/mg protein.

## Enzyme activity assays
### Superoxide dismutase (SOD)

The assay for SOD was modified from *Giannopolitis & Ries (1977)*. The activity assay was based on the ability of the enzyme to inhibit the reduction of Nitroblue tetrazolium (NBT) in a reaction mixture composed of 13 mM L-methionine, 100 μmol NBT, 0.1 mM EDTA, 16.7 μmol riboflavin and 50 mM potassium phosphate buffer (pH 7.8). The production of blue formazan, resulting from the photo-reduction of NBT, was determined by monitoring the sample absorption at 560 nm with a spectrophotometer (xMark$^{TM}$ BIO-RAD). A unit (U) of SOD was defined as the amount of enzyme required to inhibit 50% of NBT photo-reduction. The enzymatic activity was expressed in U/mg protein.

## Catalase (CAT)

CAT activity was determined following the method of *Beers & Sizer (1952)*: the reaction mixture was composed of a 25 mM $H_2O_2$ solution, 50 mM potassium phosphate buffer and 10 μL of the enzyme extract. Readings were made spectrophotometrically (xMark $^{TM}$ BIO-RAD) at 240 nm. The enzyme activity was determined by the kinetics of $H_2O_2$ degradation and expressed in U/mg protein.

## Peroxidase (POD)

Peroxidase was determined by the procedure described by *Sadasivam & Manickam (1996)* with minor modifications. Guaiacol was used as substrate. The assay was performed using 50 mM phosphate buffer, a 20 mM guaiacol solution, and a 25 mM $H_2O_2$ solution. In a 96-well microplate (Microtiter $^{TM}$), 300, 5, and 10 μL of the above solutions were placed, respectively, and, finally, 10 μL of the enzyme extract was added. The absorbance was read spectrophotometrically (xMark $^{TM}$ BIO-RAD) at 436 nm. The reading of the reaction started when the reaction absorbance was 0.05 and stopped when it reached an absorbance of 0.1. The enzymatic activity was determined by the production level of tetraguaiacol. The results were expressed in U/mg protein.

## Polyphenol oxidase (PPO)

PPO activity was measured according to the method described by *Mayer, Harel & Ben-Shaul (1966)* with minor modifications. In this case, catechol was the substrate of the enzyme. 50 mM phosphate buffer, pH 7, and a 0.1 M catechol solution were used. In a 96-well microplate (Microtiter $^{TM}$) 150 μL of the buffer, 20 μL catechol, and 20 μL of the sample were placed for each sample. Absorbance was read at 495 nm at intervals for 3 min. The specific enzymatic activity was determined by the kinetics of quinone production. The activity was expressed in U/mg protein.

## Phenylalanine ammonia lyase (PAL)

The assay for PAL was modified from *Beaudoin-Eagan & Thorpe (1985)*. Three solutions were used: a 0.5 M Tris–HCl buffer, pH 8, one of 10 mM L-phenylalanine, and one of 5 M HCl. For the reaction, 250 μL of phenylalanine solution, 125 μL of distilled water, 500 μL of the buffer, and 125 μL of the enzyme extract were added. Absorbance was spectrophotometrically measured in a 300 μL 96-well microplate (Microtiter $^{TM}$) at 290

nm. The mixture was then incubated at 37 °C in a thermostatic bath for one hour, after this time, 100 μL HCl were added to stop the reaction and the absorbance was again measured at the same wavelength. The specific activity of the enzyme was determined by the kinetics of the trans-Cinnamic acid production and expressed in U/mg protein.

## Total phenols

Total phenols were determined according to *Mng'omba, du Toit & Akinnifesi (2008)*. The stem samples (0.05 g) were homogenized in pre-chilled pestle and mortar. one mL of ice-cold methanol-acetone-water solution was added (7:7:1) to the homogenate and centrifuged at 10,000 g at 4 °C for 4 min. The supernatant was used for the quantification of total phenols performed by using 50 μL of the supernatant and adding 200 μL of distilled water and 250 μL of Folin-Ciocalteau reagent. The mixture was shaken at 800 rpm for 3 min. Then, 500 μL of a 7.5% (w/v) $NaCO_3$ solution were added. The mixture was homogenized for 1 min at 800 rpm and incubated for 15 min at 45 °C in a thermostatic shaker. Absorbance was measured in a 300 μL 96-well microplate (Microtiter $^{TM}$) at 760 nm. The concentration of phenols was expressed as mEq gallic acid/mg of sample.

## Statistical analysis

The grafting success rate, the enzyme activities, and total phenols were evaluated by analysis of variance (ANOVA) and significance among treatments was analyzed by Least Significant Differences (LSD) test at the 5% level ($p < 0.05$). Data were analyzed using R statistical software (3.5.1).

To test the relationship among enzyme activities and total phenols based on graft combination and days after grafting, the Pearson correlation was used. The correlation coefficient ranged in value from −1 to +1. An absolute value of 1 indicated a strong correlation. To test the significance of differences in enzyme activities, total phenols and graft combination, independent sample $t$-tests were conducted with the significance level of 0.05 ($p < 0.05$).

To analyze the influence of *B. subtilis* on the enzyme activity, total phenols and the graft success rate, Principal component analysis (PCA) was performed. R software (3.5.1) was used to plot the PCA map of tomato plants 15 DAG.

# RESULTS

The bacterial antioxidant activity was studied using free radical scavenging and a ferric reducing power assay. The tests were performed on eight strains (data not shown). The following results were obtained for strain BMB 44 which was the best performing strain.

## In vitro antioxidant activity of *B. subtilis*
### Resistance to hydrogen peroxide ($H_2O_2$)

In Fig. 1, the effect of $H_2O_2$ on the growth of the *B. subtilis* strain BMB 44 is shown. The results showed that all concentrations reached their maximum OD after 18 h. The highest OD, 1.6, corresponded to the control. However, despite the increase of $H_2O_2$

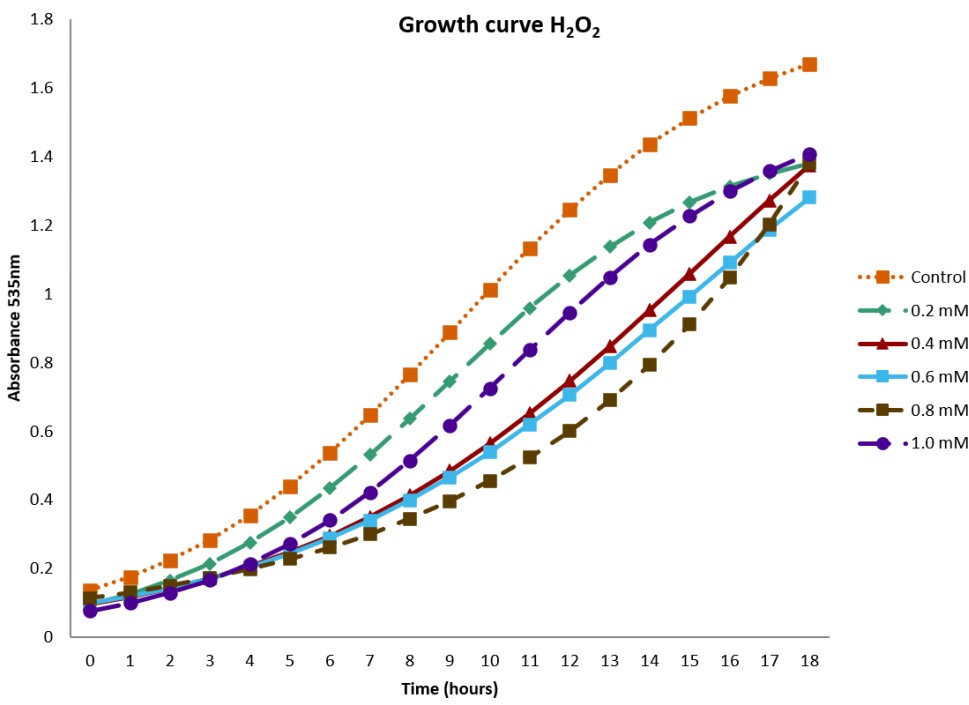

**Figure 1** Growth curve of *Bacillus subtilis* strain BMB 44 cells at different concentrations of hydrogen peroxide ($H_2O_2$).

concentrations, the lowest OD registered was 1.2. Surprisingly, at the highest concentration of $H_2O_2$, an OD of 1.4 was measured.

### Hydroxyl radical scavenging activity (OH·)

The scavenging activity for hydroxyl radicals of the strain BMB 44 of *B. subtilis* is shown in Fig. 2A. It was observed that the increase in the scavenging activity was directly proportional to the concentration of the cells. At 2.5 mL of cells at $10^6$ CFU/mL, there was a 37% scavenging rate while the lowest percentage was found in the control with a 5% scavenging ability.

### Total antioxidant activity (DPPH free radical scavenging activity)

The *B. subtilis*, strain BMB 44, was also checked for its DPPH reducing capability. The DPPH free radical scavenging activity was measured by the reduction of stable DPPH radical to non-radical DPPH-H. The scavenging activity was highly dependent and directly proportional to the concentration of cells (Fig. 2B). The highest inhibition activity was found at 2.5 mL ($10^6$ CFU/mL) with 100% inhibition but, even at a lower concentration (0.5 mL), *B. subtilis* showed about 30% of scavenging activity.

### Effect of B. subtilis on tomato grafting

To confirm the ability of *B. subtilis* to colonize the tomato grafted plants, a preliminary experiment was conducted. In all examined tissues of the inoculated plants, *B. subtilis* was observed (data not shown).

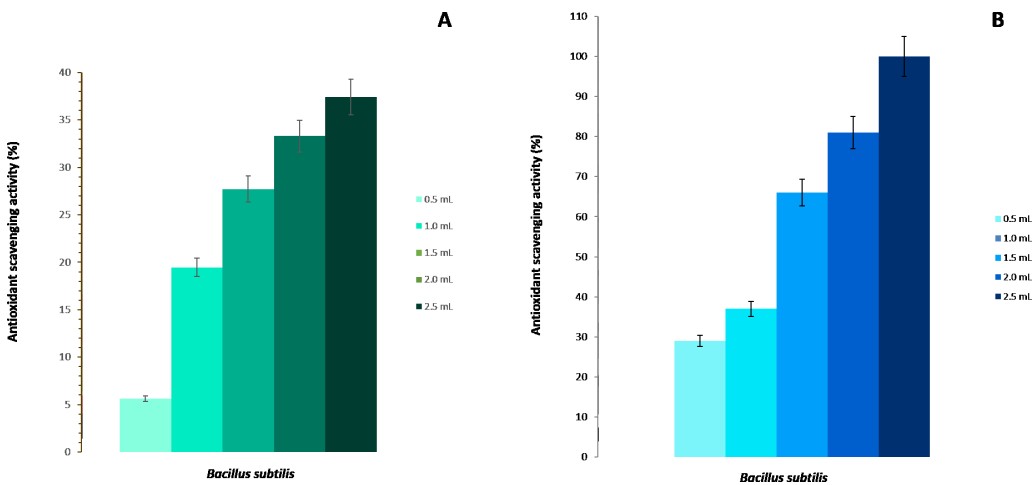

**Figure 2** **Antioxidant activity of *Bacillus subtilis* strain BMB 44.** (A) Scavenging activity on hydroxyl radicals. (B) Scavenging activity on DPPH free radicals.

**Table 1** **Effect of *B. subtilis* (strain BMB 44) application on the callus development (15 DAG) and grafting success rate (28 DAG) of "Rio Grande" grafted on "Cherry" (RGCh) and "Rio Grande" grafted on eggplant (RGBer).**

| Scion/Rootstock combination | Grafts with developed callus % | | Grafting success % | |
|---|---|---|---|---|
| | Control | *B. subtilis* | Control | *B. subtilis* |
| | 15 DAG | | 28 DAG | |
| RGCh | 74**b** | 86**a** | 99**a** | 100**a** |
| RGBer | 70**b** | 80**a** | 90**b** | 95**a** |

**Notes.**
DAG, days after grafting.
$P < 0.05$.

## Grafting success

In Table 1, the effect of *B. subtilis* on tomato-grafting success rate is reported as percentage of grafted plants that had developed a complete callus. Fifteen days after grafting, 86% of the inoculated RGCh plants already showed a complete callus development while in the water-treated plants used as control, complete callus formation was 74%. In the case of RGBer, 80% of the inoculated plants showed a complete development of the callus while only 70% of the control grafted plants developed a complete callus.

On day 28, the grafting success rate, evaluated as the result of the overall grafting procedure in inoculated and non-inoculated plants, was not significantly different for the bacterized and control RGCh plants, while for RGBer, the graft success rate of the inoculated plants was significantly higher (95%) than for the control plants (90%).

## Enzyme activities and total phenols

In this study the variations of antioxidant enzyme activities were assessed in relation to oxidative stress produced in the tissues when a plant is grafted: the variation in activity of

**Table 2 Effect of *B. subtilis* (strain BMB 44) application on enzyme (SOD, CAT) activities at 1 day after grafting (DAG), 15 DAG and 28 DAG in "Rio Grande" grafted on "Cherry" (RGCh) and "Rio Grande" grafted on eggplant (RGBer).** For each day, the different letter means significant statistical differences ($p < 0.05$) between control (water treatment) and inoculated grafts as evaluated by LSD test.

| Scion/Rootstock combination | SOD (U/mg of protein) | | | | | |
|---|---|---|---|---|---|---|
| | 1 DAG | | 15 DAG | | 28 DAG | |
| | Control | *B. subtilis* | Control | *B. subtilis* | Control | *B. subtilis* |
| RGCh | 197.8 ± 1.1**a** | 146.1 ± 3.4**b** | 302.9 ± 3.0**a** | 64.3 ± 1.9**b** | 71.2 ± 5.9**b** | 117.9 ± 4.4**a** |
| RGBer | 75.9 ± 0.5**b** | 115.9 ± 0.6**a** | 70.5 ± 1.9**b** | 76.7 ± 1.8**a** | 90.1 ± 1.1**a** | 53.4 ± 3.2**b** |
| | CAT (U/mg of protein) | | | | | |
| | 1 DAG | | 15 DAG | | 28 DAG | |
| | Control | *B. subtilis* | Control | *B. subtilis* | Control | *B. subtilis* |
| RGCh | 883.6 ± 17.0**a** | 212.2 ± 2.4**b** | 37.5 ± 0.4**a** | 17.02 ± 0.2**b** | 6.9 ± 0.1**b** | 36.5 ± 0.3**a** |
| RGBer | 218.7 ± 4.5**b** | 791.7 ± 34.1**a** | 7.54 ± 0.18**b** | 10.34 ± 0.27**a** | 15.5 ± 0.3**a** | 7.1 ± 0.2**b** |

the enzymes SOD, CAT, PPO, POD, and PAL were measured in the plants grafted with the different scion /rootstock combinations in relationship to the *B. subtilis* inoculation.

In Table 2 are reported the activities of SOD and CAT in grafted plants treated with *B. subtilis*. In the case of SOD, 1 day after grafting, in the RGCh combination, the compatible one, there is a lower activity (difference of 51 units) *in B. subtilis* treated plants in respect to the control. On the other hand, in the case of the semicompatible graft (RGBer) there is a significant increase of 40 units at day 1.

On day 15, the RGCh activity in the control was significantly higher in respect to the inoculated plants (difference of 238 units), while in the case of RGBer the bacterized plants showed higher activity in respect to the non-inoculated plants.

On day 28, an increase of SOD activity was observed in RGCh, while in RGBer the increase in activity in the inoculated plants in respect to the control was not observed.

The variation the in activity of enzyme CAT with the different graft combinations is reported in Table 2. On 1 day after grafting, in the RGCh combination, a higher enzyme activity can be observed in the non-inoculated plants presenting a difference of 671 units in respect to the grafted plants treated with *B. subtilis*. On the other hand, in the case of RGBer, the inoculated plants present higher enzyme activity (573 units) in respect to the control.

Fifteen days after grafting, in contrast with RGCh, the RGBer graft showed higher activity when inoculated. On day 28 after grafting, higher activity was observed in compatible grafted plants treated with the bacterium in respect to the non-inoculated plants, while, a reduction of activity was observed in grafted plants of RGBer treated with the bacterium in respect to the control.

Considering PPO, on day 1 after grafting (Table 3) the enzyme activity is higher in the case of the inoculated plants of RGCh, presenting a difference of 3.8 units, while in the other cases there is only a slight tendency to increase the activity of 0.1 and 0.3 units, respectively, on the control plants of RGBer. On day 15, the activity of the non-inoculated grafted plants was higher for the RGBer combination, while the bacterized plants presented the highest

**Table 3** Effect of *B. subtilis* (strain BMB 44) application on enzyme (PPO, POD) activities at 1 day after grafting (DAG), 15 DAG and 28 DAG in "Rio Grande" grafted on "Cherry" (RGCh) and "Rio Grande" grafted on eggplant (RGBer). For each day, the different letter means significant statistical differences ($p < 0.05$) between control (water treatment) and inoculated grafts as evaluated by LSD test.

| Scion/Rootstock combinations | PPO (U/mg of protein) | | | | | |
|---|---|---|---|---|---|---|
| | 1 DAG | | 15 DAG | | 28 DAG | |
| | Control | *B. subtilis* | Control | *B. subtilis* | Control | *B. subtilis* |
| RGCh | 0.6 ± 0.04**b** | 4.4 ± 0.3**a** | 4.1 ± 0.1**b** | 6.4 ± 0.02**a** | 2.1 ± 0.02**a** | 1.2 ± 0.03**b** |
| RGBer | 0.58 ± 0.015**a** | 0.52 ± 0.01**b** | 2.54 ± 0.05**a** | 1.96 ± 0.18**b** | 4.3 ± 0.3**a** | 2.77 ± 0.23**b** |
| | POD (U/mg of protein) | | | | | |
| | 1 DAG | | 15 DAG | | 28 DAG | |
| | Control | *B. subtilis* | Control | *B. subtilis* | Control | *B. subtilis* |
| RGCh | 13.54 ± 0.3**a** | 12.73 ± 0.4**a** | 34.4 ± 0.9**a** | 24.8 ± 0.5**b** | 8.3 ± 0.2**b** | 16.7 ± 0.1**a** |
| RGBer | 5.5 ± 0.2 **b** | 52.0 ± 1.5**a** | 18.4 ± 0.74**a** | 11.1 ± 0.14**b** | 6.9 ± 0.3**b** | 20.2 ± 0.2**a** |

activity for the compatible graft and the lowest activity for the RGBer combination. On day 28, the activity is slightly higher in the control grafts.

The variation in the activity of POD with the different graft combinations is reported in Table 3. On day 1 after grafting, the greatest difference in activity was found in the RGBer combination where the bacterized plants present a higher activity of more than 47 units for the control. The POD activity for RGCh is similar in the inoculated and non-inoculated plants. On day 15, RGCh and RGBer grafted plants had the highest activity when non-inoculated, while the inoculated plants showed higher activity for the control only when comparing the compatible graft. On day 28 the RGCh and RGBer inoculated plants had higher activity.

Concerning the enzyme PAL (Table 4), on one day after grafting, regardless of the graft combination, lower activity in bacterized grafted plants in respect to the controls was observed. The control in the RGCh graft showed the highest difference (7.2 units) in respect to the inoculated graft, while the semicompatible combinations have a difference of 3.8 units.

On day 15, the non-inoculated grafted plants presented the highest activity, regardless of the combination. On day 28, the control grafted plants of RGBer combinations present a slightly higher activity than the inoculated plants but with a difference of only 0.5, while no difference was found in RGCh.

In the case of total phenols (Table 4), on one day after grafting, the non-inoculated plants presented higher content of phenols for both RGCh and RGBer; on day 15, a greater content of phenols was measured in the controls for the semicompatible (6.15 units) grafts for the bacterized grafts. On the other hand, in the case of the inoculated grafts the phenol content for RGCh presented a higher content (7.6 units). The opposite pattern was observed on day 28 where control RGCh had higher phenol content respect to inoculated grafts, and RGBer control had a lower phenol content.

The overall result of inoculating *B. subtilis* is shown in Fig. S1. The enzyme activity (or total phenol content) shown was calculated by subtracting inoculated plants enzyme

**Table 4 Effect of *B. subtilis* (strain BMB 44) application on enzyme PAL and total phenols at 1 day after grafting (DAG), 15 DAG and 28 DAG in "Rio Grande" grafted on "Cherry" (RGCh) and "Rio Grande" grafted on eggplant (RGBer).** For each day, the different letter means significant statistical differences ($p < 0.05$) between control (water treatment) and inoculated grafts as evaluated by LSD test.

| Scion/Rootstock combinations | PAL (U/mg of protein) | | | | | |
| --- | --- | --- | --- | --- | --- | --- |
| | 1 DAG | | 15 DAG | | 28 DAG | |
| | Control | *B. subtilis* | Control | *B. subtilis* | Control | *B. subtilis* |
| RGCh | 7.7 ± 0.02**a** | 0.5 ± 0.01**b** | 0.31 ± 0.006**a** | 0.22 ± 0.01**b** | 0.26 ± 0.008**a** | 0.17 ± 0.012**b** |
| RGBer | 3.9 ± 0.1**a** | 0.1 ± 0.01**b** | 0.65 ± 0.008**a** | 0.29 ± 0.016**b** | 0.9 ± 0.013**a** | 0.4 ± 0.02**b** |
| | TOTAL PHENOLS mEq of gallic acid/mg of sample | | | | | |
| | 1 DAG | | 15 DAG | | 28 DAG | |
| | Control | *B. subtilis* | Control | *B. subtilis* | Control | *B. subtilis* |
| RGCh | 15.2 ± 0.2**a** | 6.7 ± 0.5**b** | 6.07 ± 0.08**b** | 7.6 ± 0.03**a** | 8.4 ± 0.1**a** | 6.5 ± 0.1**b** |
| RGBer | 23.5 ± 0.3**a** | 7.3 ± 0.1**b** | 6.15 ± 0.12**a** | 4.05 ± 0.08**b** | 5.3 ± 0.1**b** | 7.6 ± 0.1**a** |

activity (or total phenol content) from control plants enzyme activity (or total phenol content).

### Relationship between enzyme activities and total phenols at different days after grafting

We next performed a statistical correlation analysis with respect to the graft combination on the enzyme activities and total phenols recorded at 1, 15, and 28 DAG (Fig. 3). On day 1 (Figs. 3A and 3B), for both RGCh and RGBer, CAT showed positive relationship especially with defense enzymes involved in the wound reaction such as SOD. The latter, at the same time showed a positive correlation with POD. In addition, in the case of RGCh only, CAT also correlated positively with PAL and total phenols while SOD had a strong positive relationship with PAL and total phenols. In turn, total phenols had a strong positive correlation with POD and PAL. In the case of RGBer, CAT and POD were positively correlated, in addition PPO, PAL, and total phenols also showed positive correlation. On day 15 (Figs. 3C and 3D), enzymes related to the formation of the vascular bundle were positively correlated. For RGCh some enzymes correlated positively as in day 1: CAT with SOD and PAL, and SOD with POD and PAL. Positive correlation was also observed for POD with CAT and PAL; and PPO and total phenols. In the case of RGBer, as in day 1, CAT had a positive correlation with SOD but also PPO with PAL and total phenols. In turn, total phenols had a high positive correlation with PAL. Surprisingly, for RGBer, POD showed a positive correlation with PPO, PAL and total phenols.

On day 28 (Figs. 3E and 3F), where the differentiation of the new vascular bundle has already started, for RGCh, POD and total phenols are positively correlated as in day 1. CAT and SOD, and SOD and POD keep a positive correlation as in day 1 and 15. SOD and total phenols correlate positively as it was observed in day 1, while CAT and POD also showed positive correlations as in day 15. For RGBer, CAT correlates positively with SOD as in days 1 and 15. In addition, CAT and SOD showed a positive relationship with PPO and PAL. PPO and PAL always showed positive correlation form day 1 to day 28.

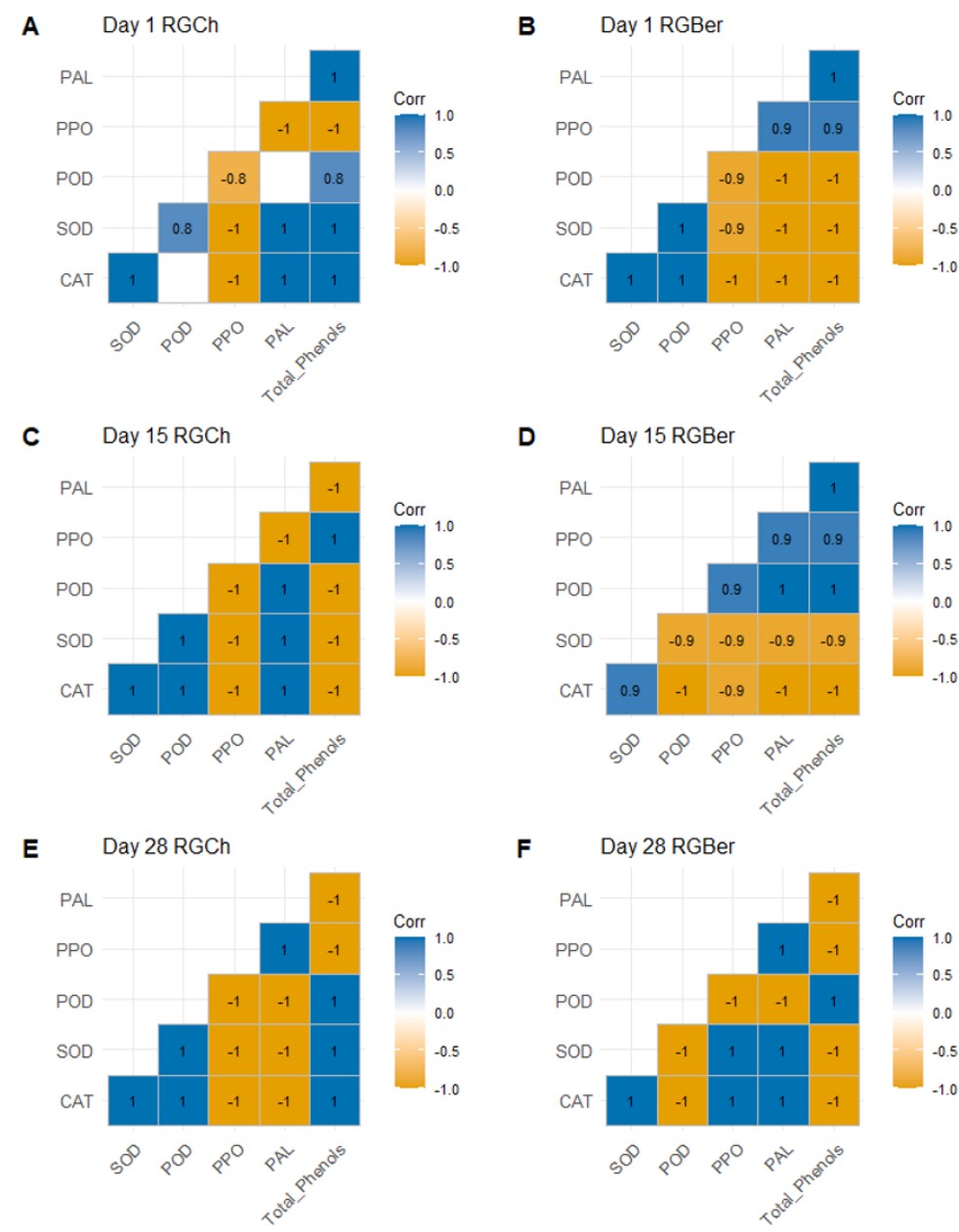

**Figure 3** Pearson correlation for enzyme activities (CAT, SOD, POD, PPO, PAL) and total phenols of graft combinations (RGCh and RGBer) investigated at 1, 15, and 28 DAG. (A) RGCh 1 DAG. (B) RGBer 1 DAG. (C) RGCh 15 DAG. (D) RGBer 15 DAG. (E) RGCh 28 DAG. (F) RGBer 28 DAG. Positive correlations are displayed in blue and negative correlations in yellow colors. The color intensities are proportional to the correlation coefficient. Correlations with $p$ value >0.05 are considered insignificant and are left in blank. CAT:Catalase, SOD:Superoxide dismutase, POD:Peroxidase, PPO:Polyphenol oxidase, PAL:Phenylalanine ammonia-lyase. RGCh: "Rio Grande" grafted on "Cherry", RGBer: "Rio Grande" grafted on eggplant. DAG: Days after grafting.

Negative relationships were also observed: on day 1 (Figs. 3A and 3B), PPO were correlated negatively with all enzymes and total phenols for RGCh combination. However, for RGBer, CAT, SOD and POD, all had negative relationship with PPO, PAL and total phenols. On day 15 (Figs. 3C and 3D), again for RGCh, PPO and total phenols had a negative relationship with CAT, SOD and POD. In addition, PAL was negatively correlated with PPO and total phenols. For RGBer, CAT and SOD were negatively correlated to POD, PPO, PAL, and total phenols. On day 28 (Figs. 3E and 3F), for RGCh, negative relationships were found for CAT, SOD, and POD in respect to PPO, and PAL, but also total phenols with PPO and PAL. However, for RGBer, total phenols were negatively correlated to all enzymes except for POD. POD showed negative correlation with SOD, CAT, PPO, and PAL.

The activities of some enzymes presented opposite relationships based on the graft combination. On day 1, for RGCh, PPO correlated negatively with PAL and total phenols, while in the case of RGBer, these were positively correlated. On day 15, PPO and POD, PAL, and PPO, and total phenols and PAL correlated negatively. The opposite happened when analyzing the RGBer combination. CAT and SOD, and CAT and PPO also showed an opposite relationship depending on the graft combination. On day 28, CAT and SOD correlated negatively with PPO and PAL while in the case of RGBer, these had a positive relationship.

To further explore the effect of *B. subtilis* on enzyme activity, total phenols, and graft success rate with the graft combination, a PCA analysis was performed using the enzyme activity, total phenol content, and graft success rate as descriptors. As shown in Fig. 4, the two principal components (PC1 and PC2) represented 83.98% of the data variance. The first component accounted for 53.1% of the variance and the second one to 30.88% of the variance.

PC1 is strongly correlated with five of the original variables (Table S1). The first principal component increases with decreasing CAT, SOD, POD, PPO, and total phenols scores but PC1 correlates most strongly with CAT and POD. Dominant variables for the second component (PC2) were PAL, PPO, SOD, and graft success. PC2 increases with increasing SOD and PAL and with decreasing PPO and graft success (Table S1).

As shown in the PCA plot (Fig. 4), the grafted plants can be grouped into four distinct clusters. Based on PC1, RGCh and RGBer controls form two single groups and inoculated RGCh and RGBer, the other two groups. Similarly, considering PC2, the graft combination also formed four single groups: RGCh (control) and RGCh (inoculated) form the first two groups and, RGBer (control) and RGBer (inoculated) the third and fourth group. Such differences are likely to be due to the enzymes that have heavy influences on PIC1 and PC2 respectively. Given that PC1 reveals the most variation, differences among clusters along PC1 are larger than PC2 differences.

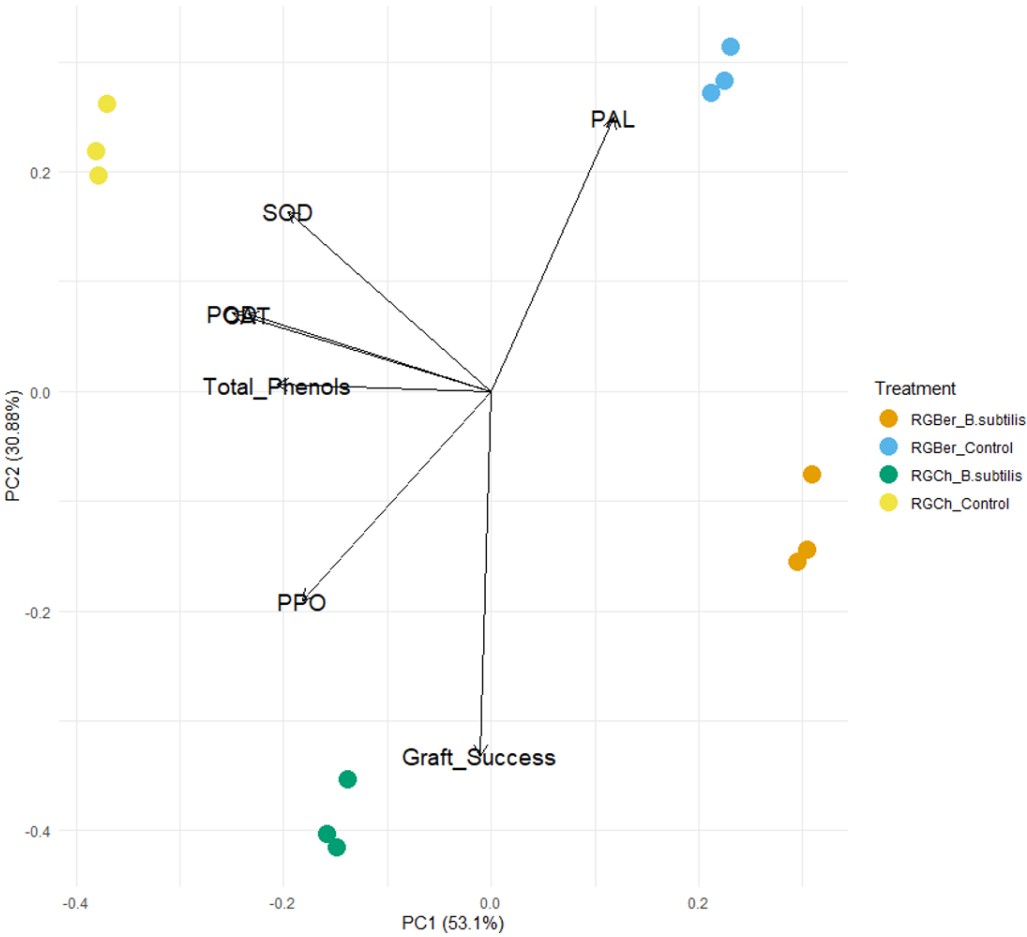

**Figure 4 Principal component analysis (PCA) of enzyme activity, total phenols and graft success of tomato grafted plants.** Plants from two different graft combinations were compared: RGCh: "Rio Grande" grafted on "Cherry", RGBer: "Rio Grande" grafted on eggplant. Enzyme activity, total phenols and graft success were measured comparing control (water treatment) and inoculated grafts. The PCA shows the first and second principal components with the explained variance in brackets.

## DISCUSSION

### In vitro antioxidant activity of *B. subtilis*

Microbial cells have several defense mechanisms. To prevent damage by ROS, organisms have evolved multiple detoxification mechanisms including various enzymatic or non-enzymatic systems (*Asada, 1994*; *Ahmad et al., 2010*). Among these enzymes, the combined action of SOD and CAT is critical in mitigating the effects of oxidative stress. They maintain the free radicals at levels that are not toxic to the cells. However, the ability of bacteria to overcome oxidative stress is related to the levels and types of antioxidant enzymes that they possess (*Amanatidou et al., 2001*; *Poole, 2012*). Several growth-promoting bacteria have been reported to possess antioxidant activity (*Han & Lee, 2005*; *Upadhyay et al., 2012*; *Kang et al., 2014*). *B. subtilis* has been extensively studied (*Hecker & Völker, 2001*) and shown to possess an adaptation mechanism against $H_2O_2$. This bacterium undergoes a

typical bacterial stress response when exposed to low concentrations (0.1 mM) of hydrogen peroxide but protection was also shown to be induced at higher concentrations (10 mM) and many proteins are induced including the scavenging enzymes, CAT (*Loewen & Switala, 1987*; *Dowds, 1994*), SOD and POD (*Mols & Abee, 2011*). Our results confirm the capacity of *B. subtilis* to react to stress conditions. At very high concentrations (1.0 mM) of hydrogen peroxide, the bacteria are unaffected by the $H_2O_2$ treatment and it is only after 18 h that it reaches a plateau. In the same way, our results confirm previous results of *Yan et al. (2006)* showing that *B. subtilis* has the capacity of scavenging radicals presenting a scavenging activity of more than 35%. In a biological system, no enzyme specifically destroys OHo. The most effective defense against OHo induced damage is to reduce the intracellular concentration of components in the Fenton reaction such as $H_2O_2$ and iron. This can be achieved by enzymes that directly breakdown $H_2O_2$ such as CAT or sequestration of transition metal and repression of iron uptake (*Hameed & Lee, 2009*). In our study, 37% scavenging activity was obtained.

In this sense, the results of DPPH antioxidant capacity measured in our study, were similar to other studies (*Kadaikunnan et al., 2015*). In our study, *B. subtilis* already scavenges 30% even at low concentrations and is capable of neutralizing 100% of the radicals at greater concentrations. This suggested that its antioxidant properties may help to reduce the level of oxidative stress associated with mechanical injuries created during grafting and different physiological stages.

## Effect of *B. subtilis* on tomato grafting

Many developmental stages can be recognized in the formation of a graft union. The early stage in herbaceous plants begins within 4 days and is characterized by the death of cell layers at the graft interface as a wound reaction (*Moore, 1984*; *Tiedemann, 1989*). After cutting, cell division leads to the formation at the wound site of undifferentiated stem-cell-like tissue called callus. The callus and the tissues surrounding the cut differentiate to phloem and xylem (*Melnyk, 2017*). According to *Fernández-García & Olmos (2004)*, the subsequent union of the newly formed vascular strand with the original vascular bundle in both rootstock and scion begins between days 4 and 8 and is fully developed after 15 days. In our study, 15 days after grafting, 86% of the inoculated RGCh plants had already fully developed callus with respect to the control (74%). In the case of RGBer, 70% of the water-treated plants had developed full callus while 80% of the bacterized plants showed full development. In both cases, RGCh and RGBer inoculated plants presented a significant higher percentage. Besides the callus formation due to the wound reaction, it is possible that *B. subtilis* could be influencing the formation of callus given by its ability to produce hormones (3-indole acetic acid and gibberellins) (*Sabir, 2013*; *Ishak et al., 2016*) but could also be due to a better condition of the tissues given by the antioxidant defense enzymes response (*Vardharajula et al., 2011*; *Rais et al., 2017*). 28 days after grafting, the graft success was not significantly different for the bacterized and control RGCh plants, while for RGBer, the graft success rate of the inoculated plants was significantly higher (95%) than for the control plants (90%). Thus, even in the case not fully compatible grafts, *B. subtilis* can positively influence the grafting results and further support previous findings showing the

benefits of plant inoculation with *B. subtilis* in promoting growth and in mitigation of abiotic and biotic stress effects (*Gajbhiye et al., 2010*; *Singh et al., 2012*).

During the developmental stages of the graft, enzymes are differently and the effect that *B. subtilis* may have on this regulation was also studied. Several studies demonstrated the benefits of inoculating bacteria in plants (*Bonaterra et al., 2003*; *Vardharajula et al., 2011*). In this study we report that plants inoculated with *B. subtilis* presented an increase in the antioxidant enzymes CAT, POD, PPO, PAL, and in total phenols levels. In another previous study, *Bacillus* spp. were also assessed to induce an increase in activity of antioxidant enzymes against *Pyricularia oryzae* (*Rais et al., 2017*). The application of *Bacillus* enhanced PPO and PAL activity but also changes in SOD and POD were observed in that study as response to the fungal infection. It has also been demonstrated that in the case of abiotic stresses, such as salinity stress, the activity of antioxidant enzymes in wheat increases with the increasing of salinity stress but plants treated with PGPR, such as *B. subtilis* and *Arthrobacter*, showed a reduction of activity of antioxidant measured enzymes as compared to uninoculated plants and among all antioxidant activities studied. The maximum reduction was recorded in CAT activity (*Upadhyay et al., 2012*). Initially, when the mechanical damage is induced in the grafted plants, there is a burst of free radicals (*Savatin et al., 2014*) and the antioxidant machinery activates. Later on, when the graft union has been reestablished, the lignification processes may intervene (*Aloni et al., 2008*).

Superoxide dismutase is an important antioxidant enzyme and constitutes the first level of defense against superoxide radicals in plants. SOD catalyzes the dismutation of $O_2^-$ to $H_2O_2$ and $O_2$. Although exposing plants to stress situations, such as grafting, would trigger the antioxidant defense systems, there are indications that in incompatible rootstock/scion combination either the level of reactive oxygen species can be increased or decreased if a less efficient detoxification system is initiated (*Aloni et al., 2008*; *Nocito et al., 2010*).

Our results indicate of the response in tomato grafting when comparing the two grafting combinations with different compatibility. During the following days the tendency in both cases is to diminish but the bacterized plants keep the units of SOD even lower.

The highest level of CAT activity was observed in the compatible graft on day 1 after grafting, while the opposite response was observed in the semicompatible graft. These results confirm findings of *Fernández-García & Olmos (2004)* who reported that in tomato, catalase is the enzyme more involved in the cell controlling process of $H_2O_2$ production that takes place after grafting. In addition, the most noticeable effect of *B. subtilis* was seen on day 1 in the compatible graft where the activity is considerably reduced in respect to the control.

It is known that genes encoding for the enzymes like PAL, PPO, and POD are developmentally and tissue-specifically regulated and may be induced by environmental stresses (*Pina & Errea, 2008*). PAL is generally recognized as a marker of environmental stress and an important step in the pathway regulating the synthesis of flavonoid compounds, xylogenesis, and formation of lignin, one of the main cell wall polymers (*Rogers & Campbell, 2004*). *Pina & Errea (2008)* demonstrated for the first time that the enhancement of the level of PAL transcription results in an accumulation of phenol and our observations on the two graft combinations, are consistent with the above-mentioned

studies. In the case of the bacterized grafted plants enzyme activity is always lower than in the controls suggesting that, in the case of the inoculated plants, the control of the stressful conditions by *B. subtilis* could have reduced the activity of PAL.

POD is also reported as an important antioxidant enzyme involved in stress response by previous studies (*Has-Schön et al., 2005*; *Rajeswari & Paliwal, 2008*). Assuming that grafting is a relevant stress factor for herbaceous plants, the increase of peroxidase activity following grafting may explain this idea. In our study, it was observed that there was an increase in POD activity 15 days after grafting. Some researchers also reported that in tomato grafts, peroxidase activity increased day by day after the graft (*Fernández-García & Olmos, 2004*). Similarly, in another study, peroxidase activity increase was found in melon 14 and 24 days after grafting (*Aloni et al., 2008*). Some researchers suggested that different graft combinations give different reactions to grafting (*Feucht, Schmid & Christ, 1983*; *Hudina et al., 2014*; *Pina & Errea, 2005*). The POD activity is higher for bacterized plants on days 1 and 28 for both graft combinations. This response suggests that *B. subtilis* may have a radical scavenging effect as a reaction to wounding and later as a reaction to lignification. POD is also a catalyzer in the polyphenol biosynthesis and, together with PPO, is responsible for the production of phenolic compounds which contribute to the reinforcement of cell barriers and therefore they confer resistance against diseases. In addition, they are involved in the wounding stress response (*Gainza, Opazo & Muñoz, 2015*; *Saltveit, 2015*). Recent data demonstrate that several biochemical pathways are affected during graft union formation (*Koepke & Dhingra, 2013*). One of these is the metabolism of phenolic compounds (*Mng'omba, du Toit & Akinnifesi, 2008*). As expected in a normal wound reaction, an intense production of new phenolic compounds has been reported during the establishment of a graft union (*Tiedemann, 1989*; *Hartmann et al., 2002*). Phenolic compounds are uncommon in bacteria, but their accumulation is a distinctive characteristic of plant response to stress. Our results show higher total phenol content for both control graft combinations on day 1, even though on the following days the content in either control or bacterized plants varied but was kept lower in respect to day 1. This response may be due to the nature of the scion and rootstock itself. The lower phenol content could be due to the antioxidant effect of *B. subtilis* which takes the plant to a lower stress condition.

PPO physiologically has an important role in plant defense and is also involved in the lignification of plant cells. This could explain the peaks that can be observed after 15 days in the RGCh control combination. The inoculated plants have higher activity than in the control in the compatible grafts, but the activity tends to decrease in the case of the semicompatible grafts. This response could be also related to modulation of the response by *B. subtilis*.

In the present research, the activity was enhanced or reduced depending on the enzyme, the time when the activity was measured, and the graft combination. In general, *B. subtilis* decreased the activity of SOD, CAT, and PAL as well as the quantity of total phenols, on day 1 on the compatible grafts. In the case of the semicompatible grafts, the activity of PAL, PPO, and the total phenols quantity was decreased. On day 28, CAT, PAL, and PPO showed reduced activity for RGCh but in the case of RGBer, the SOD, CAT, and PAL

showed reduced activity as well as the total phenols. *Krishna, Kumar & Kumar (2011)* also tested *B. subtilis* for antioxidant activity by enzymatic and non-enzymatic parameters and changes of antioxidant activity were observed. Moreover, in the last stage, the grafted plants should have their vascular connections formed, and, therefore, the enzymatic activity could change accordingly to the graft union formation.

Taken together, the above results, showed that the mechanical damage, such as the one caused by wounding in grafted plants, generates ROS. In the present study, the activity of the measured antioxidant enzymes SOD, CAT, PAL, PPO, and POD in the graft union of different graft combinations treated with strain BMB 44 of *B. subtilis* was significantly reduced or increased as compared to control plants (non-inoculated). The most evident effect can be noticed indeed on day 1 where for the SOD, CAT, and PAL enzymes the activity was significantly decreased while it was elicited for the PPO and there was not a significant change in POD. This can be attributed to the ability of bacteria to limit producing ROS through modulation of the enzymatic defense system by increasing or decreasing antioxidant enzyme activities according to the physiological stage of the graft and the compatibility level of the graft combination.

Our results, which apparently seem in disagreement with the premise that stress resistance in plants is related to more effective antioxidant systems (*Li et al., 2008*; *Bianco & Defez, 2009*) is an outcome of the same positive effect and indicate that inoculated grafted plants felt less stress as compared to non-inoculated plants (*Omar et al., 2009*).

To better describe and quantify the association among the enzymes and between the enzymes and total phenols, the Pearson's correlation was used (Fig. 3). The production of ROS is involved in defense processes of plants and it is a general event following grafting. Hydrogen peroxide is an important component of the plant response (*Orozco-Cárdenas & Ryan, 2002*). Given its toxic nature, hydrogen peroxide works as a signal molecule which activates complex mechanisms and production of ROS-scavenging enzymes. Peroxidases have a role in detoxification of ROS (*Minibayeva, Beckett & Kranner, 2014*) but CAT and SOD are also considered antioxidant markers in plants (*Demidchik, 2015*). Thus, according to the available literature, the positive relation of CAT, SOD and POD on day 1 after grafting, suggests that antioxidant mechanisms were activated in both RGCh and RGBer plants given the reaction to wounding. Moreover, it has also been documented that total phenols, PAL, PPO, and POD enzymes increase in wounded tissues (*Ngadze et al., 2012*) and in some cases these could be a sign of graft incompatibility (*Pina & Errea, 2009*).

Therefore, PAL, PPO, and POD are the defense related enzymes associated with phenolic and lignin production in plants facing biotic and abiotic stress(es) (*Mandal, Chakraborty & Dey, 2010*). PAL is the key enzyme in the synthesis of phenol and lignin. It is the first and most important enzyme in the phenylpropanoid pathway. PPO and POD participate in the cell wall polysaccharide processes and catalyze the oxidation of phenols to quinones. These are also involved in the suberization and lignification of plant cells during the defense reaction. In our study, for RGCh, on day 1 after grafting, PAL correlated positively with POD and total phenols; and for RGBer combination, PAL increased as PPO and total phenols increased.

A previous study (*Fernández-García, Carvajal & Olmos, 2004*) suggested that in tomato grafted plants, vascular bundles are fully formed 15 days after grafting and that peroxidase and catalase are closely related. This could explain why the strong positive relationship is kept for CAT and SOD and at the same time SOD increases as POD and PAL increases. For RGBer grafts, surprisingly, the enzymes related to the phenolic and lignin production kept a positive relationship as it is shown by observing POD with PPO, and PAL and total phenols. This suggests that vascular reconnection may still be active (*Mellerowicz et al., 2001*). In fact, on day 28, for both RGCh and RGBer, PAL and PPO, and PPO and total phenols had a strong positive correlation.

Strong negative correlations were present for both combinations regardless of the time. On day 1 and 15, in the case of RGBer, PPO, PAL and total phenols correlated negatively with CAT. On day 28, again CAT had a negative relationship with POD and total phenols. On the other hand, RGCh showed negative correlation for PPO and CAT on day 1. On day 15, CAT was negatively correlated with PPO and total phenols, and on day 28 with PPO and PAL. The enzyme SOD also had an inverse relationship with POD, PPO, PAL or total phenols depending on the graft combination and day, suggesting that the different types of antioxidative profiling may be due to the different developmental stages of the grafted plant.

The present analysis revealed that the activities of CAT, SOD, POD, PPO, PAL, and total phenol content undergo a differential modulation (increasing or decreasing) upon graft combination and time after grafting. Nevertheless, the data indicates that, regardless of the graft combination and days after grafting, the enzymes SOD and CAT always kept a strong positive relationship confirming these two enzymes as the first line defense antioxidants (*Racchi, 2013*).

In this study, PCA analysis was performed to analyze which of the measured enzymes was more important and the influence of the total phenols on the inoculated e non-inoculated compatible and semicompatible grafted plants. PCA analysis revealed that the first two principal components accounted for 83.98% of variability.

It should be noted that there exists an inverse relationship between CAT, SOD, POD, PPO, and total phenols with the first component (PC1) (Table S1). This suggests that *B. subtilis* might be affecting the enzyme and total phenols at this stage (15 DAG) which could be important for graft success. The dominant variables for PC2 were PAL PPO, SOD and graft success. However, SOD and PAL are directly related to PC2 while PPO and graft success are inversely related to the second principal component.

The inoculated plants are influenced by *B. subtilis*. As it is shown in Fig. 4, if the PC1 is considered, grafted plants form separated clusters, two regarding the inoculated plants and two regarding the control plants. On the contrary, respect to PC2 four clusters can also be observed but with in respect to the graft combination, RGCh or RGBer. This confirms that also compatibility confers specific characteristics based on the graft combination.

## CONCLUSIONS

*Bacillus subtilis* strain BMB 44, was shown, through *in vitro* evaluation, to have high antioxidant capacity. The in vivo application of this strain on grafting tomato plants

showed its relevant effect on the modulation of enzyme activities and total phenols level immediately after grafting when there is an outburst of free radicals as well as in the other stages of the graft recovery period when the oxidative stress can be associated with the reconnection of the vascular tissue. Moreover, it was observed that the capacity of the bacterium of lowering or increasing the enzyme activity and total phenols level in plants also depends on whether graft combination is compatible or semicompatible.

The results of this study suggest that *B. subtilis,* acting on the modulation of the antioxidant response, may represent a useful tool for mitigation of the adverse effect of grafting enhancing graft success and survival rate.

## ACKNOWLEDGEMENTS

Horacio Claudio Morales Torres for his help in the statistical analysis. Emanuel de Jesús Guzman Valdez and Anita Camacho Ramírez for technical assistance.

### Funding

Maria D. Arias Padró's work was supported by a doctoral scholarship from Consejo Nacional de Ciencia y Tecnología (CONACYT, Mexico). The funders had no role in study design, data collection and analysis, decision to publish, or preparation of the manuscript.

### Grant Disclosures

The following grant information was disclosed by the authors:
Consejo Nacional de Ciencia y Tecnología (CONACYT, Mexico).

### Competing Interests

The authors declare there are no competing interests.

### Author Contributions

- Maria D. Arias Padró conceived and designed the experiments, performed the experiments, analyzed the data, prepared figures and/or tables, authored or reviewed drafts of the paper, and approved the final draft.
- Emilia Caboni analyzed the data, prepared figures and/or tables, authored or reviewed drafts of the paper, and approved the final draft.
- Karla Azucena Salazar Morin conceived and designed the experiments, performed the experiments, analyzed the data, prepared figures and/or tables, and approved the final draft.
- Marco Antonio Meraz Mercado analyzed the data, prepared figures and/or tables, and approved the final draft.
- Víctor Olalde-Portugal conceived and designed the experiments, analyzed the data, prepared figures and/or tables, authored or reviewed drafts of the paper, and approved the final draft.

## Data Availability

Raw data are available in the Supplemental Files.

## Supplemental Information

Supplemental information for this article can be found online at http://dx.doi.org/10.7717/peerj.10984#supplemental-information.

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
