# Peer review of "Effect of Bacillus subtilis on antioxidant enzyme activities in tomato grafting"

_PeerJ, doi:10.7717/peerj.10984_

## Round 0.1 · original submission · Major Revisions

- Why were these tomato and cucumber cultivars selected and used for grafting in this study? Provide reasons.

- What was your working hypothesis?

- Data were poorly analyzed. Need more robust data analysis.

- Please check several minor edits in the manuscript.

Reviewer 1 ·

Basic reporting

No comment

Experimental design

• It was not clearly stated how the grafted plant sample was processed to obtain the enzyme extract for measuring enzyme activity and measuring phenol quantity.

Validity of the findings

• There was so much variation in enzyme activities between control and Bacillus treated grafted plants according to the scion/root stock combination as well as time of sampling, that no conclusion regarding the effect of Bacillus treatments affecting the free radical scavenging activity on the grafted samples can be drawn.

• Frequently, authors make erroneous or misleading statements such as,
- In line 37 for Abstract, It is stated that “For all combinations, the quantity of total phenols decreased”. But in Table 3. For Day 1 in RGPep, Day15 in RGCh and Day 28 in RG1ber, total phenol was significantly increased in Bacillus treatment.
- In line 477-8 in discussion, it is stated that “In general, B. subtilis decreased the activity of SOD, CAT, POD and PAL as well as the quantity of total phenol, on day 1 on the compatible grafts”. But Table 2 shows no significant difference in POD activity between control and Bacillus treatment.

• It is not stated/ discussed whether these different enzyme activities originated from plants or bacteria or whether Bacillus is successfully colonized in the grafted plants or whether Bacillus treatment promotes the grafting performance and plant growth.

Reviewer 2 ·

Basic reporting

The study is innovative and clear technically. They expressed and presented well.
You took three rootstocks, third one is totally different and practically it has no use.

Experimental design

Experimental design is well illustrated.

Validity of the findings

Work is novel and useful data they have generated which would be quite beneficial for the future studies.

Additional comments

This needs some English improvement. Otherwise, Its a good study. I would recommend for publication as full paper.

Reviewer 3 ·

Basic reporting

English needs improvement, especially with spelling and punctuations. I corrected many spelling errors and placed commas throughout, but not in whole document. authors are advised to learn from my suggestions and make similar changes.

nevertheless, the article was well researched and the results were well presented.

Experimental design

no comment

Validity of the findings

conclusions are well stated and referenced (compared) to prior research

Additional comments

please review and study my comments and make necessary corrections

Annotated reviews are not available for download in order to protect the identity of reviewers who chose to remain anonymous.

Reviewer 4 ·

Basic reporting

The structure is well represented by professional data, tables, and figures. Sufficient literature used with enough background justification. But the usage of English is very poor esp sentence formation. Many typographical errors, improper use of spacings, superscript/subscript, etc. Needs to revise English in many parts esp in the discussion.

Experimental design

The objective of the research is clear with a proper hypothesis. The experimental design is appropriate. The investigation is okay, but it would be valid it is provided with the grafting success rate in the treatments at the end of the experiment based on the visual observation.

Validity of the findings

Finds are well presented with appropriate statistical analysis. Again, it would be great if it is supported by the grafting success percentage.

Additional comments

Kindly revise the article with proper English and rectify the errors.

Line 46: Cucurbitaceae and Solanaceae
Poor writing style and unnecessary spacing: eg: line 97, 104, 159, 179, 220, etc. in using spaces. 269-280, unnecessary spacing between paragraphs: Line 300, 311.
Improper usage for superscript and subscript eg: Line 107: H2O2, Line 159: 117umol s-1 m-2
Line 139: Kindly give the tray size and the dimensions used for the experiment.
Line 140: What is the media composition used to raise the seedlings?
Line 140: Which part of the scion was treated? or the entire scion? It was done before the cut or after the cut? Not clear.
Line 153: What is the height of the rootstock and scion with how many leaves?
Line 159: 117umol s-1 m-2) to 117 umol: Spacing and superscript (s–1m–2).
Line 94, 241: Unnecessary italics.
In many places letter “O” was replaced with the number “0” in H202 Vs. H2O2. Eg: Line 363, 365.
Rewrite the results with better English with clear sentences.
Kindly justify why you take three different species of the rootstock. (Two Solanaceous and one Cucurbitaceous).
It is worth to mention the grafting success percentage of graft union at the final stage.
Kindly check the references and use punctuations appropriately eg: full stop at the end of the references. Some have and some missing. Also, different styles used. Kindly change it to the journal style. & is missing in many references before the last author (line:588, 569). Some references instead of &, “and” are used (line 616).

Fig.1.: Provide space before unit nm.

Annotated reviews are not available for download in order to protect the identity of reviewers who chose to remain anonymous.

---

## Round 0.2 · accepted · Accept

Dear Dr. Olalde-Portugal,

Thank you for revising your submission to PeerJ.

I am writing to inform you that your manuscript - Effect of Bacillus subtilis on antioxidant enzyme activities in tomato grafting - has been Accepted for publication. Congratulations!

Best regards,

Tika Adhikari